# Selective Segmental Pulmonary Angiography: Anatomical, Technical and Safety Aspects of a Must-Learn Technique in Times of Balloon Pulmonary Angioplasty for Chronic Thromboembolic Pulmonary Hypertension

**DOI:** 10.3390/jcm10153358

**Published:** 2021-07-29

**Authors:** Maite Velázquez, Nicolás Maneiro, Ana Lareo, Agustín Albarrán, Sergio Huertas, Allende P. Olazábal, Juan F. Delgado, Sergio Alonso, Fernando Sarnago, Julio García Tejada, Fernando Arribas, Pilar Escribano

**Affiliations:** 1Cardiology Department, University Hospital 12 de Octubre, 28041 Madrid, Spain; nicolasmmaneiro@gmail.com (N.M.); lareo.ana@gmail.com (A.L.); agustin.albarrang@hotmail.com (A.A.); shuertas89@gmail.com (S.H.); allendeolazabal_30@hotmail.com (A.P.O.); juan.delgado@salud.madrid.org (J.F.D.); fernando.sarnago@salud.madrid.org (F.S.); julio.garcia@salud.madrid.org (J.G.T.); fernando.arribas@salud.madrid.org (F.A.); pilar.escribano.subias@gmail.com (P.E.); 2Instituto de Investigación Sanitaria Hospital 12 de Octubre (imas12), 28041 Madrid, Spain; sacharterina@salud.madrid.org; 3Centro de Investigación Biomédica en Red de Enfermedades Cardiovasculares (CIBERCV), 28041 Madrid, Spain; 4Facultad de Medicina, Universidad Complutense de Madrid (UCM), 28041 Madrid, Spain; 5Radiology Department, University Hospital 12 de Octubre, 28041 Madrid, Spain

**Keywords:** selective pulmonary angiography, chronic thromboembolic pulmonary hypertension, balloon pulmonary angioplasty, pulmonary arteries’ anatomy

## Abstract

With the advent of balloon pulmonary angioplasty (BPA) for non-surgical chronic thromboembolic pulmonary hypertension (CTEPH) patients, there is renewed interest in the pulmonary angiography technique. This technique is still the standard imaging modality to confirm CTEPH, which, in addition, helps to determine the most appropriate treatment. Furthermore, learning this technique fulfills two main purposes: to identify BPA candidates and to provide the operator with the catheter handling needed to perform BPA. Operators interested in performing BPA must learn not only the pulmonary arteries’ anatomy, but also which are the best angiographic projections and the most suitable catheters to canalize and display each segmental branch. Unfortunately, this information is scarce in the literature. With this goal, learning the diagnostic pulmonary angiography technique can be a first step on the way to perform BPA. Although there are descriptions on how to perform a pulmonary angiography with balloon-tipped catheters and the digital subtraction technique, this technique does not provide operators with the catheter knowledge and manual skill needed to cannulate each segmental branch. In contrast, learning the conventional selective segmental pulmonary angiography (SSPA) technique provides the operator with this knowledge and skills. In this review, based on the experience of the authors, we describe the pulmonary arteries’ anatomy and detail the practical aspects of the SSPA procedure, with the aim of providing operators with the anatomical and technical knowledge needed to perform BPA. We also summarize the contemporary complications of SSPA in CTEPH patients at a reference center.

## 1. Introduction

The advent of balloon pulmonary angioplasty (BPA) for patients with non-surgical chronic thromboembolic pulmonary hypertension (CTEPH) has raised interest in the invasive pulmonary angiography technique [1]. Although digital subtraction angiography (DSA) has been considered the gold standard for characterizing vessel morphology in CTEPH [2], it is being challenged by advances in non-invasive modalities like computed tomography pulmonary angiography (CTPA). Both techniques can identify surgically accessible chronic clots [3,4]. However, more selective segmental angiography, cone-beam CT and ECG-gated area detector CT may be better for pre-BPA planning by providing greater resolution than conventional DSA. The problem in daily clinical practice is that some of these imaging techniques are not widely available and require expertise [5]. Learning how to perform a diagnostic SSPA provides the operator with the knowledge of the pulmonary arteries’ anatomy and the ability to selectively cannulate each segmental branch, essential skills to ultimately perform BPA. Unfortunately, there is little information in the literature on how to properly perform a SSPA. Furthermore, data on the risks and complications of SSPA in patients with severe pulmonary hypertension (PH) are scarce. 

In this review, we describe in detail the pulmonary arteries’ anatomy, provide recommendations regarding the practical aspects of the SSPA technique and report contemporary procedural complications in patients with severe PH due to CTEPH in an expert center. The pulmonary artery (PA) anatomy description and the technical aspects described below are based on our experience of 452 SSPA cases and of 450 BPA cases performed by the cath-lab members of a referral CTEPH center.

## 2. Pulmonary Arteries’ Anatomy 

The main pulmonary artery (PA) arises from the pulmonary conus of the right ventricle, takes a posteromedial direction and, below the aortic arch, bifurcates into the right and left main pulmonary arteries at the level of the carina [6,7]. After bifurcation, the left pulmonary artery (LPA) arises as a straight continuation of the main PA, while the right pulmonary artery (RPA) arises with a marked angle to the right. Below, we describe the most common pattern of all lobar and segmental branches of both pulmonary arteries (Figure 1 and Figure 2), from the point of view of an interventional cardiologist, with the aim of teaching how to selectively cannulate all of them. The figures shown contain all the patterns that can appear in the pulmonary arteries of CTEPH patients, such as webs, bands, slits, diffuse distal narrowing, pouch terminations and total occlusions. 

### 2.1. Lobar and Segmental Arteries of the Upper Lobes

Both upper lobes have three segmental arteries: apical (A1), posterior (A2) and anterior (A3). The most common patterns are as follows: (1) an apico-posterior trunk (A1/2) plus an anterior segmental branch (A3) (Figure 3 and Figure 4) and (2) an apico-anterior trunk (A1/3) plus a posterior segmental branch (A2) (Figure 5). In the right lung there is a right upper lobe artery originating the segmental branches (Figure 6). However, in the left lung there is not a left upper lobe artery as such; thus, the trunk/segmental branches arise directly from the LPA (Figure 3 and Figure 5). There are anatomical variants: three segmental arteries can originate independently in the LPA; the posterior segmental artery (A2) can arise directly from the right interlobar artery (Figure 7) and, very rarely, the posterior segment of the upper lobe is irrigated by branches from the superior segmental artery (A6) of the lower lobe (Figure 8). 

### 2.2. Lobar and Segmental Arteries of the Lingula and the Middle Lobe

The segment of the LPA and RPA comprised between the branches of the upper lobes and the branches of the middle lobe or the lingula is called the interlobar artery (Figure 9). In the left lung, the interlobar artery originates the lingula artery, with an anterior direction, and the superior segmental artery of the left lower lobe (LLL), opposite to it, with a posterior direction (Figure 10). The lingula artery bifurcates early in two segmental branches—cranial (A4) and caudal (A5) (Figure 11)—which sometimes have an independent origin in the interlobar artery. 

Once the lingula artery has exited, the interlobar artery is called the LLL artery. In the right lung, the middle lobe artery arises from the anteromedial wall of the interlobar artery, with an anterior direction, before the interlobar artery turns downward. It divides soon in two segmental branches: medial (A5) and lateral (A4) (Figure 12). Both segmental arteries can also arise independently directly from the interlobar artery (Figure 13). 

After the exit of the middle lobe artery, the interlobar artery becomes the right lower lobe (RLL) artery. The superior segmental artery of the RLL (A6) arises opposite to the middle lobe artery, with a posterior direction (Figure 14). 

### 2.3. Lobar and Segmental Arteries of the Lower Lobes

Both lower lobes have five segmental arteries: four for the basal pyramid and one for the apical segment of the lower lobe. Commonly, the four basal segmental branches originate from two trunks. In the left lung, there is a posteromedial trunk (A7/10) and an anterolateral trunk (A8/9) (Figure 15). In the right lung, there is an anteromedial trunk (A7/8) and a posterolateral trunk (A9/10) (Figure 16). Each trunk originates two segmental branches. The artery for the apical (or superior) segment (A6) has an independent origin in the interlobar artery, before the basal trunks’ exit (Figure 17). In the left lung, the superior segmental artery (A6) bifurcates in two subsegmental branches—cranial and caudal (Figure 18)—which sometimes arise independently in the LLL artery. Less commonly, some segmental basal branches have an independent origin in the lower lobe artery (Figure 19).

## 3. Angiographic Projections 

All segmental arteries should be filmed in two orthogonal views, mainly lateral and anteroposterior (AP) projections. The lateral projection avoids overlapping of the middle lobe or lingula branches with those of the lower lobes [6]. Many times, segmental branches with an anterior distribution are better visualized using cranial and oblique angulations instead of the AP projection. We recommend a cranial right anterior oblique (RAO) view for the anterior segmental artery (A3) of the left upper lobe, the lingula branches and the anterior segmental branches (A8) of the LLL, and a cranial left anterior oblique (LAO) view for the anterior segmental artery of the right upper lobe (A6), the middle lobe artery (Figure 20) and the anterior segmental branches (A8) of the RLL. 

## 4. Procedure

In our center, the SSPA is performed in the cath-lab, on an outpatient basis, awake, with mild oral or IV sedation and after 8 h of fasting. We stop vitamin K antagonists and administer low-molecular-weight heparin (LMWH) at an anticoagulant dose 48 h before the procedure, giving the last LMWH dose the night before the procedure. If patients are under direct-acting oral anticoagulants, they do not receive the last scheduled dose previous to the procedure. After the procedure, the oral anticoagulant therapy is resumed the same day. We perform right heart catheterization (RHC) through a 7 French introducer prior to the SSPA, to settle the patient’s risk status. The non-ionic iso-osmolar contrast agent iodixanol (Visipaque 270; GE Healthcare) is routinely used. 

## 5. Venous Access

In our experience, the femoral veins allow for an easier and more efficient catheter manipulation to canalize all the segmental branches, being less bothersome for the patient. It is the venous access of 92% cases of our series. The jugular access or a forearm vein force the operator to be closer to the X-ray generator. Thus, we consider them second choices. Having a vena cava filter does not preclude a femoral access. Any catheter (balloon-tipped or not) can be carefully advanced through the filter, assuring the guidewire is running ahead and under fluoroscopy, when advancing or withdrawing the catheter, to avoid the filter mobilization or deterioration. Nine patients of our cohort had a vena cava filter. In all of them, the SSPA was done through the femoral venous access crossing the filter, and none of them were damaged. 

## 6. Catheter Selection and Positioning

Most of the segmental pulmonary arteries can be selectively canalized with a multipurpose (MP) diagnostic catheter. The MP catheter can have two bends: an A-bend (hockey stick shape with a 120 degree bend) and a B-bend (gradual 90 degree bend). Both can have one end-hole (MP-A1 or B1, respectively) or two side-holes plus one end-hole (MP-A2 or B2, respectively) (Figure 21). The MP-A2 is the most useful for performing SSPA. Its side-holes allow for high-contrast injection flow rates with a low risk of iatrogenic vessel injury. 

If a RHC is done before the SSPA, the MP-A2 is placed in the PA exchanging the Swan-Ganz catheter with a 0.025–0.035 inches exchange guidewire. If we do not perform a RHC first, or if the position of the balloon-tipped catheter in the PA is lost, the PA can be easily accessed from the femoral vein with the MP-A catheter, with the maneuver described in Figure 22. 

After crossing the tricuspid valve in the AP view, the MP catheter is retrieved to a middle position in the chest and rotated counterclockwise. Its tip will point towards the right ventricle (RV) outlet tract, overlapping with the backbone. A 0.035 inches guidewire advanced through it easily reaches the LPA, which is a continuation of the main PA. We then cannulate all the LPA segmental branches with the following catheters:
Left upper lobe. The apico-posterior trunk (A1/2) is easily cannulated in the AP view. We should rotate the MP-A2 catheter counterclockwise in the proximal segment of the LPA to make the tip face upwards (Figure 23). The apical and posterior segmental arteries (A1 and A2) are well visualized by filming in the AP and lateral views. The anterior segmental artery (A3) is engaged more easily with the Judkins left catheter (JL) 3.5 or 4. Position the JL below the aortic arc in the AP view with the distal tip facing the left lateral chest wall. Rotate the catheter counterclockwise and withdraw it slightly. The distal tip of the catheter will climb 2–3 cm, engaging the anterior segmental branch (A3) (Figure 24). Film this branch in the cranial RAO and lateral views.Lingula. The lingula artery is easily cannulated in the AP view with the JL 3.5/4 catheter (depending on the PA dilatation). The maneuver and position in the AP view is the same as described for the anterior segmental artery of the left upper lobe (A3), though somewhat lower (Figure 25). Amplatz left 1 or 2 is also a good choice for catheterization of the lingula branches. In addition, this catheter gives extra support when performing BPA of these branches. The two lingula segmental arteries are well displayed in the lateral and in the cranial RAO/cranial AP views.Left lower lobe. The two trunks of the basal pyramid and the superior segmental artery (A6) are easily cannulated with the MP-A2 in the lateral projection (Figure 26). The anterolateral trunk (A8/9) sometimes requires the Judkins right (JR) 4 or the JL 3.5/4. We should sequentially film both trunks and the superior segmental artery (A6) in the lateral and AP view.

Next, we switch to the right lung with the JL 3.5/4, not with the MP catheter. The double bend of the JL catheter makes this exchange very easy, saving time and radiation. To do this, we should place the JL in the main PA with the tip pointing towards the RPA in the AP view. Advancing the 0.035 inches guidewire, the catheter distal bend will open, and advancing everything as a block, including the catheter and the guidewire, the catheter will enter into the RPA (Figure 27). Once in the RPA, we exchange the JL by the MP-A2 to canalize the right lung segmental branches.
Right upper lobe. The right upper lobar artery arises about 10 cm from the bifurcation. It is cannulated in the AP view with the tip of the MP-A2 facing upwards. The apico-posterior segmental artery (A1/2) is canalized just by advancing the MP distally once in the right upper lobe artery. The AP and lateral views display the apical (A1) and posterior branches (A2) (Figure 28). The anterior segmental artery (A3) is canalized in the lateral view, rotating the MP counterclockwise from its position in the apico-posterior artery (A1/2). With this maneuver, the tip of the catheter moves downwards, facing the sternum and engaging the anterior segmental artery (A3) (Figure 29). This branch should be filmed in lateral and cranial AP/cranial LAO view.Middle lobe. The middle lobe artery originates in front of the superior segmental artery of the RLL (A6), with an anterior direction, as does the lingula in the left lung (Figure 14). It is canalized with the MP, in the lateral view. The distal angulated segment of the MP should be bent, pressing against the bifurcation of the RLL, and rotated counterclockwise, with the tip of the catheter pointing towards the sternum (Figure 30). The JR 4 is also a good alternative. This artery bifurcates early in two segmental branches: lateral (A4) and medial (A5). The lateral and cranial LAO views display both branches. If these segmental branches have an independent origin in the RPA, they should be cannulated and filmed independently (Figure 31).Right lower lobe. The lateral projection and the MP are the best choices to canalize the two trunks of the basal pyramid and the superior segmental artery (A6). Use the AP and lateral views to assess the four segmental basal branches (Figure 32) and the lateral projection to display the superior segmental artery (A6) (Figure 16 and Figure 17).

Our recommendation, therefore, to assess all the segmental arteries of both lungs is to begin with the left lung, canalizing with the MP-A2 catheter the branches of the LLL and the apico-posterior trunk of the left upper lobe (A1/2). Then, switch to the JL 3.5/4 to canalize those arteries with an anterior origin: the lingula artery, the anterior segmental artery of the left upper lobe (A3) and, sometimes, the anterolateral trunk of the LLL (A8/9). Subsequently, switch to the RPA with the JL and exchange again to the MP-A2 with which, in most patients, it will be possible to cannulate all the right lung segmental branches. The JR 4 will help to canulate any branch not seen. In Figure 1 and Figure 2, we detail each segmental branch and the most appropriate catheter to canalize them.

## 7. Contrast Injection Settings 

The lateral holes of the six French MP-A2 diagnostic catheters allow for high flow and high volume injections with the power injector, with a low risk of damaging the artery wall. If diagnostic JL or JR catheters are used, we recommend to make them lateral holes with an intramuscular needle. 

Usually, 8 mL at 4 mL/s or 6 mL at 3 mL/s is enough to assess most segmental branches. The trunks of the basal pyramid (which originate two segmental branches) require more contrast: 12 mL at 6 mL/s or 15 mL at 8 mL/s. A small contrast test prior to pump injection will rule out a super-selective position in a small branch (risk of rupture and hemoptysis). Maximum pressure is typically set at 600 psi. Use the reverse lateral to reduce the radiation dose to the operator (Figure 33). 

## 8. Differential Characteristics between Diagnostic Selective Segmental Pulmonary Angiography Technique and Central Pulmonary Angiography with Digital Subtraction

Digital subtraction angiography (DSA) has long been considered the gold standard to characterize the involvement of the pulmonary arteries in CTEPH [2,8,9,10]. In recent years, it is progressively being replaced by CTPA for the diagnosis and assessment of operability [9]. However, a normal CTPA does not rule out CTEPH, since distal segmental or subsegmental involvement may go unnoticed [8,9]. In fact, the publication on “the state of the art in CTEPH” after the last World Symposium in PH in 2018, establishes that, due to the advent of BPA as a therapeutic option in HPTEC, an accurate assessment of very distal vessels of the pulmonary tree is needed. Moreover, it also says that conventional DSA may not always be suitable for providing fine details and that a more selective segmental angiography may be better for pre-BPA planning by providing greater resolution, particularly in the more distal vessels [5]. Unfortunately, SSAP is not widely available. Expertise is required to perform it, but this is not easy to acquire, as there are no descriptions in the literature on how to perform it.

A recent publication by the San Diego group, a pioneer in CTEPH TEA surgery, describes their DSA technique to perform quality diagnostic pulmonary angiography. They perform it with a Berman catheter, through jugular access, by a single injection in each main pulmonary artery during deep inspiration, with biplane acquisition and the digital subtraction technique post-process [2]. This image acquisition requires a big radiologic biplane flat detector to include the entire lung in the field of view, as well as digital subtraction technology and image post-processing. However, as in many institutions, the interventional cardiology units have assumed CTEPH diagnosis and BPA treatment; thus, a different imaging approach in this center is needed. Most interventional cardiology cath-labs have small cardiac detectors. Interventional cardiologists are not familiar with the digital subtraction technique nor is this software usually available in their cath-labs. Therefore, the San Diego’s group DSA pulmonary angiography technique is not applicable in many interventional cardiology units [2]. Additionally, the central injection pulmonary angiography may not give, in some cases, the image quality needed to accurately discriminate whether or not there is very distal involvement [5], while SSPA may. Finally, contrast injection through a balloon-tipped catheter in each main pulmonary artery with diagnostic purposes does not provide novel operators with the handling of catheters required to independently canalize each segmental branch, which is needed to perform BPA. For these reasons, we propose that interventional cardiologists or interventional radiologists who want to perform BPA should first of all learn to carry out SSPA diagnostic procedures, following the technique explained in this manuscript. It is a very precise technique, like DSA central angiography, used to accurately analyze the distal pulmonary arteries involvement. It is more widely available in cardiology cath-labs and it facilitates operators to perform BPA once the diagnostic SSPA procedure has been learned. Thankfully, imaging technology keeps improving. Some groups use C-arm CT (CACT) to visualize CTEPH peripheral lesions in detail and to guide BPA [11]. This technique, which uses cross-sectional 3D imaging with high special resolution, is able to identify the target lesions for BPA in the forefront of the procedure and to visualize them on the display in the angiographic suite. Thus, CACT guidance in BPA may help to avoid guidewire perforations, dissection and rupture of target vessels and, thereby, increase the patient safety. Although it is available nowadays in a minority of cardiologic cath-labs, hopefully these 3D graphic representations of the segmental and subsegmental arteries will help in the next future to safely perform BPA.

Our PH Multidisciplinary Unit began in 1996, we have been a National Reference Center for HP since 2015 and we have belonged to the PH European Reference Network (ERN) since 2018. Our unit started the TEA program in 2000 and the BPA program in May 2013, having performed 287 TEA and 450 BPA procedures up until December 2019. The experience our unit has gained in performing SSPA is shown in this systematized description of the technique. We think that SSPA can be a qualified diagnostic procedure alternative to DSA pulmonary angiography for interventional cardiologists interested in the field of diagnosis and BPA treatment of CTEPH patients. Through the SSPA technique described in this manuscript, learning how to selectively cannulate each segmental artery and to know the projections that best allow for their respective evaluation fulfills two very important issues in the management of CTEPH: (1) to have the best pulmonary arteries’ images to select patients subsidiary for BPA and (2) to acquire the knowledge of the pulmonary vascular tree anatomy and the ability to move inside it to be able to perform BPA.

## 9. Selective Segmental Pulmonary Angiography Complications

Any invasive procedure performed in the cath-lab can be associated with fatal complications related to vascular puncture and contrast administration. Furthermore, the hemodynamic fragility of patients with severe PH increases the risk of any procedure. There are few references in the literature regarding the safety of the pulmonary angiography technique, and those that exist are old [12]. Hofmann et al. showed that the mortality rate of pulmonary angiography in a 202 patients single-center series with severe PH was 1.5% [13]. The Hoeper’s multicenter registry (1214 pulmonary angiographies) describes seven major adverse events; one was a death secondary to diffuse pulmonary hemorrhage and electromechanical dissociation (mortality rate 0.08%) [12]. Although they do not specify the pulmonary angiography technique, such a low death rate can be explained by the fact that all the centers were expert PH centers. We analyzed baseline and procedural characteristics of our series of 452 SSPA (Table 1) as well as technique-related complications (Table 2). There were complications in 22 procedures (4.9%), most were mild and related to bleeding at the puncture site (Table 2). No complications appeared 4 days after the procedure. One patient with severe PH and right ventricular disfunction died in the cath-lab due to hemodynamic instability and refractory respiratory failure. The PH severity, the right ventricular disfunction, the volume overload and the delay in recognizing the hemodynamic impairment probably played an important role in the fatal outcome. Therefore, although operators must be aware of the fragility of these patients, our peri-procedural mortality rate of 0.2% confirms the safety of the SSPA technique in an expert PH center.

## 10. Conclusions

Learning the diagnostic SSPA technique is, in some patients, crucial to establish if a patient with distal CTEPH is a good candidate for BPA. It is also one of the ways to acquire the anatomic knowledge and manual skills to be able to approach this percutaneous therapy. Our work describes in detail the technique for performing diagnostic SSPA in CTEPH patients based on the performance of 452 SSPA and 450 BPA procedures in a national PH referral center. Currently, this diagnostic procedure is safe in patients with severe PH when performed in an expert center, with a periprocedural mortality rate of 0.2%.

## Figures and Tables

**Figure 1 jcm-10-03358-f001:**
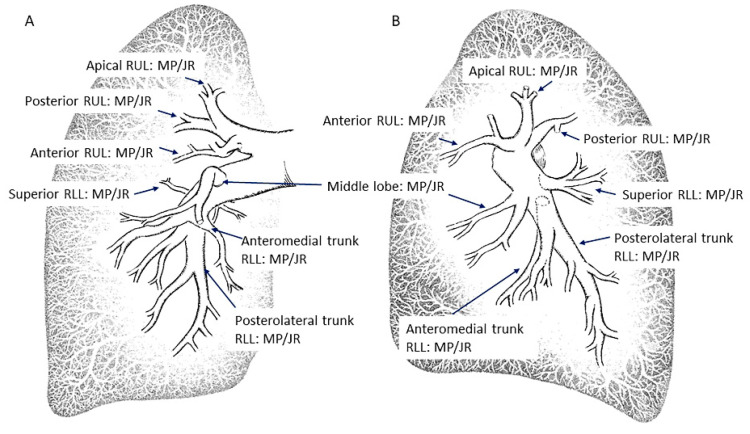
Right lung segmental arteries and catheters to canalize them: (**A**) anteroposterior view and (**B**) lateral view. JR: Judkins right; MP: multipurpose; RUL: right upper lobe; RLL: right lower lobe.

**Figure 2 jcm-10-03358-f002:**
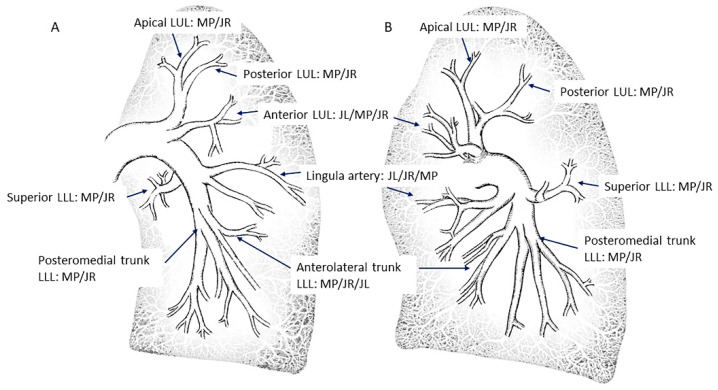
Left lung segmental arteries and catheters to canalize them: (**A**) anteroposterior view and (**B**) lateral view. LUL: left upper lobe; LLL: left lower lobe. Abbreviations as in Figure 1.

**Figure 3 jcm-10-03358-f003:**
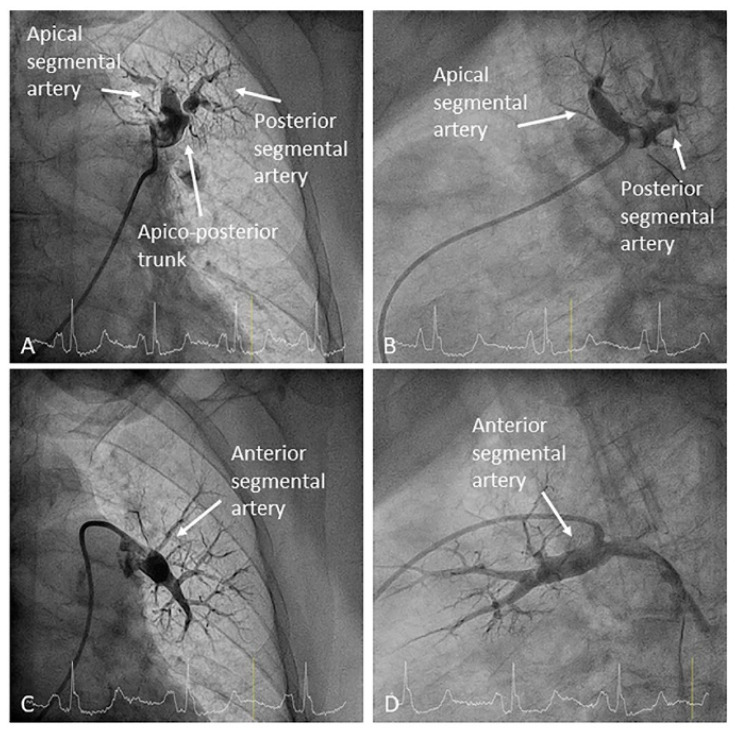
Left upper lobe arteries: (**A**,**B**) apico-posterior trunk (A1/2) and (**C**,**D**) anterior segmental branch (A3). (**A**,**C**) Anteroposterior view and (**B**,**D**) lateral view.

**Figure 4 jcm-10-03358-f004:**
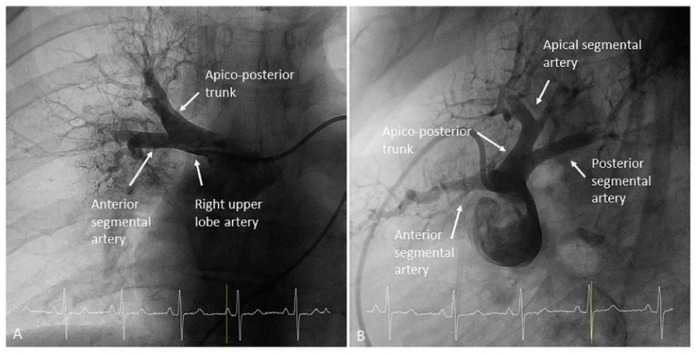
Right upper lobe arteries: apico-posterior trunk (A1/2) and anterior segmental branch (A3). (**A**) Anteroposterior view and (**B**) lateral view.

**Figure 5 jcm-10-03358-f005:**
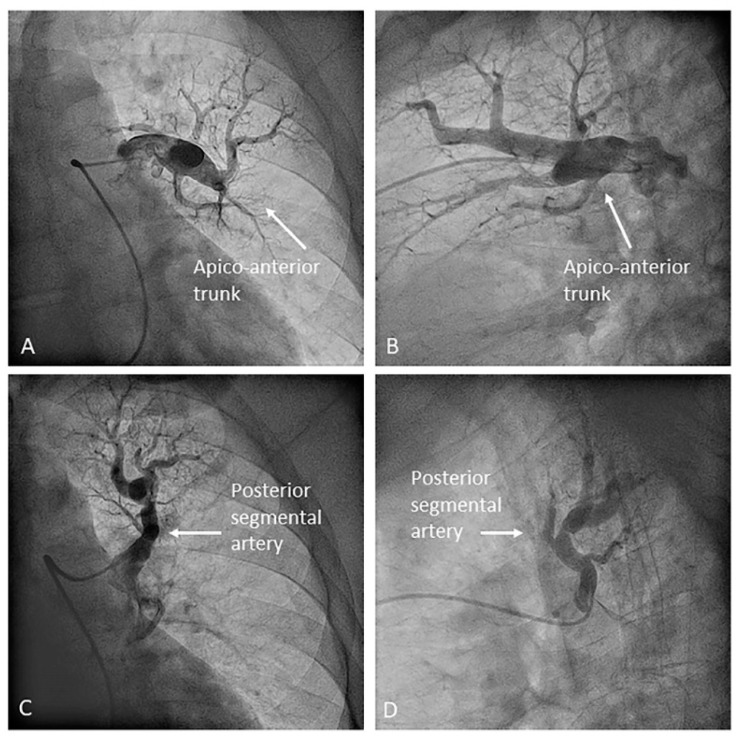
Left upper lobe arteries: (**A**,**B**) apico-anterior trunk (A1/3) and (**C**,**D**) posterior segmental branch (A2). (**A**,**C**) Anteroposterior view and (**B**,**D**) lateral view.

**Figure 6 jcm-10-03358-f006:**
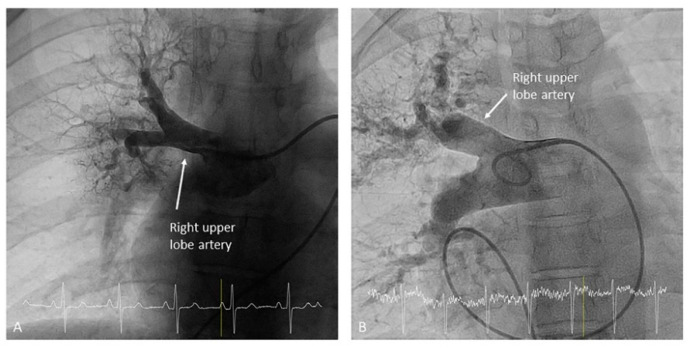
Right upper lobe artery ((**A**,**B**): anteroposterior view).

**Figure 7 jcm-10-03358-f007:**
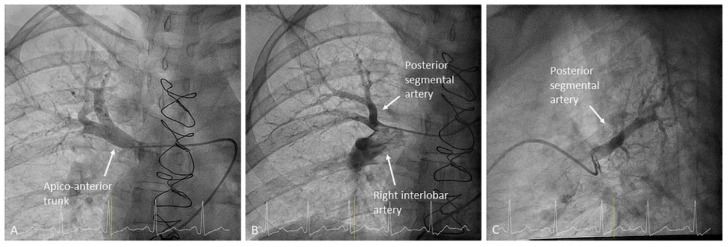
Right upper lobe branches: (**A**) apico-anterior trunk (A1/3) and (**B**,**C**) posterior segmental artery (A2) with the origin in the right interlobar artery. (**A**,**B**) Anteroposterior view and (**C**) lateral view.

**Figure 8 jcm-10-03358-f008:**
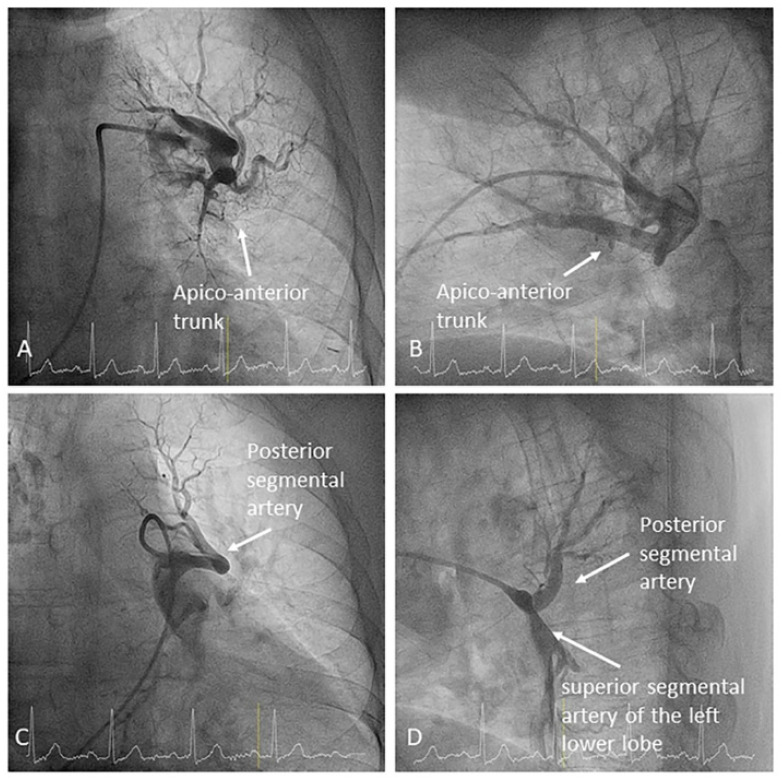
Left upper lobe branches. The posterior segmental artery (A2) originates in the superior segmental artery (A6) of the left lower lobe: (**A**,**C**) anteroposterior view and (**B**,**D**) lateral view.

**Figure 9 jcm-10-03358-f009:**
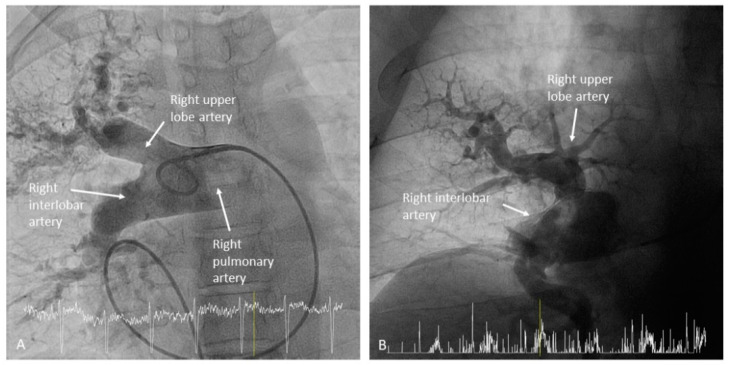
Right interlobar artery ((**A**,**B**): anteroposterior view).

**Figure 10 jcm-10-03358-f010:**
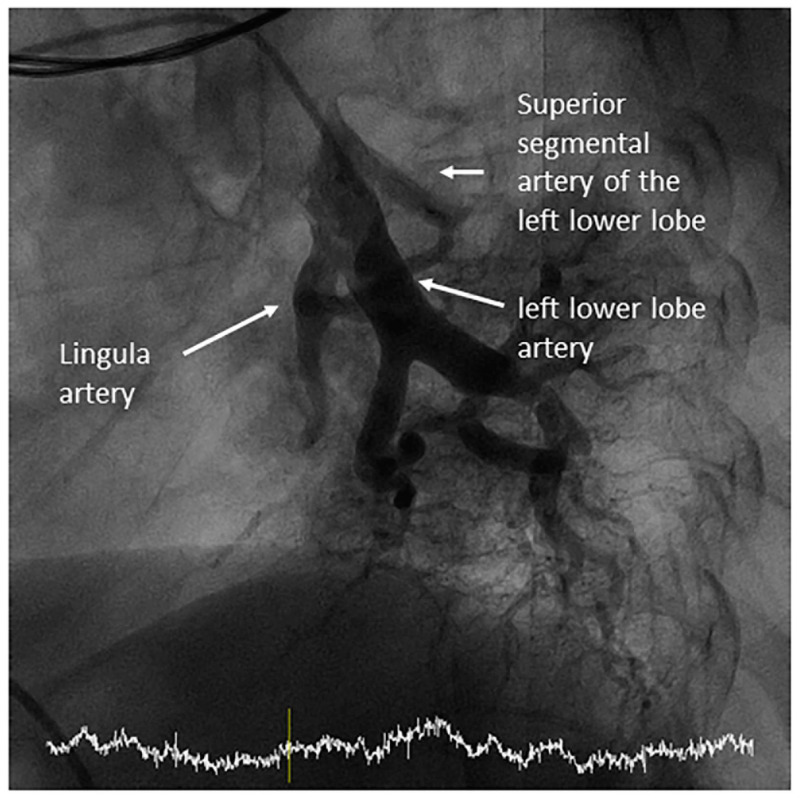
Left pulmonary artery branches (lateral view).

**Figure 11 jcm-10-03358-f011:**
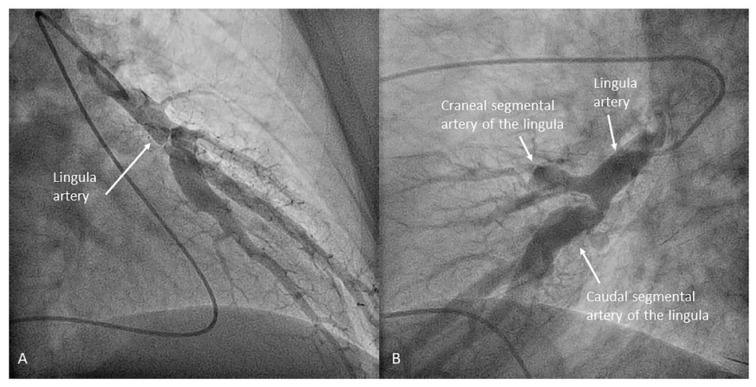
Lingula artery and segmental branches: cranial (A4) and caudal (A5): (**A**) anteroposterior view and (**B**) lateral view.

**Figure 12 jcm-10-03358-f012:**
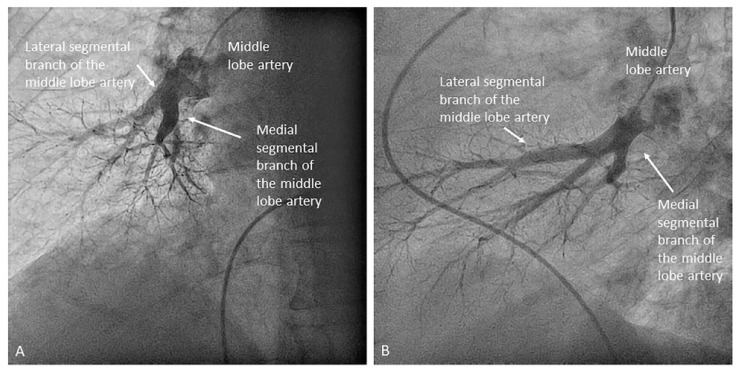
Middle lobe artery and segmental branches: medial (A5) and lateral (A4): (**A**) anteroposterior view and (**B**) lateral view.

**Figure 13 jcm-10-03358-f013:**
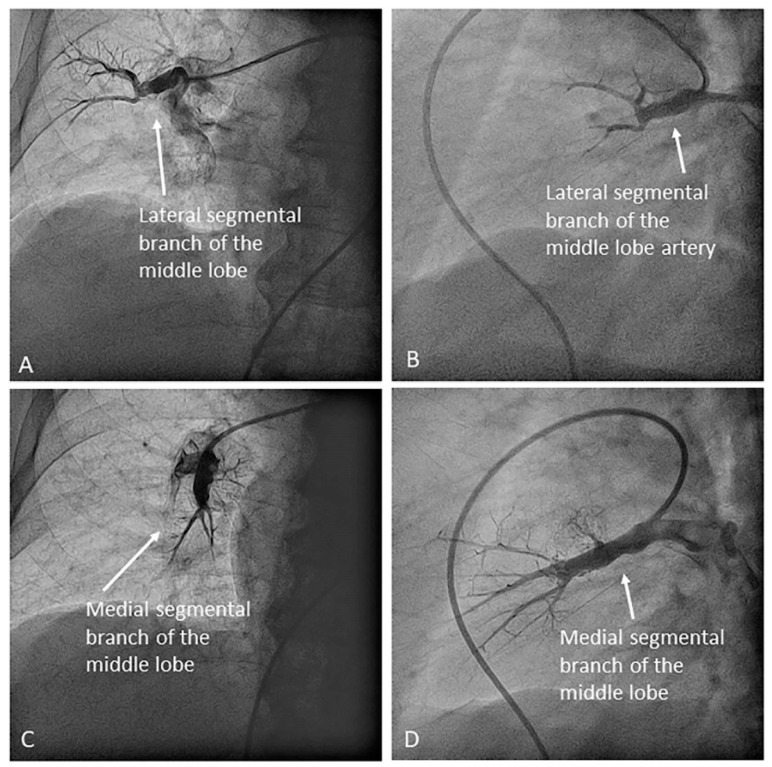
Independent origin of the middle lobe segmental branches in the right interlobar artery: (**A**,**C**) anteroposterior view and (**B**,**D**) lateral view.

**Figure 14 jcm-10-03358-f014:**
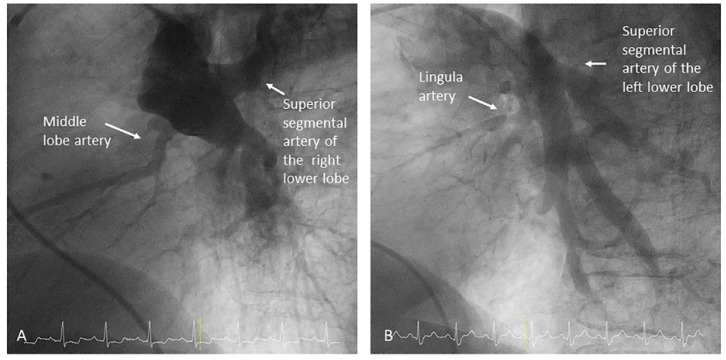
Lower lobes superior segmental arteries (A6), opposite to (**A**) the middle lobe and (**B**) lingula arteries. ((**A**,**B**): lateral view).

**Figure 15 jcm-10-03358-f015:**
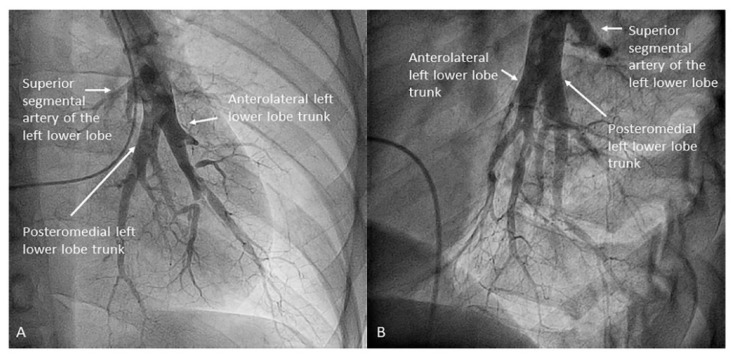
Branches for the left lower lobe: (**A**) anteroposterior view and (**B**) lateral view.

**Figure 16 jcm-10-03358-f016:**
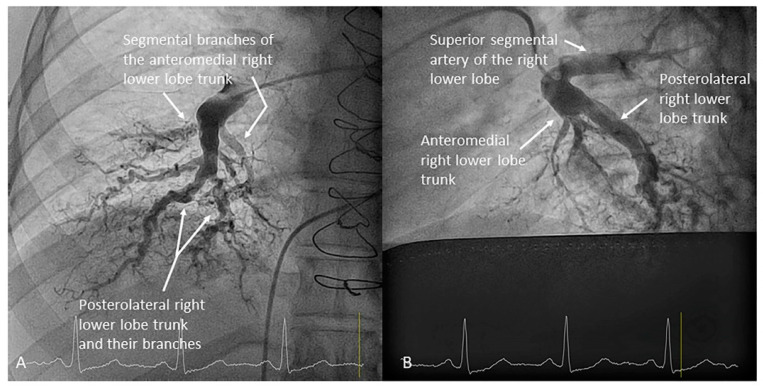
Branches for the right lower lobe: (**A**) anteroposterior view and (**B**) lateral view.

**Figure 17 jcm-10-03358-f017:**
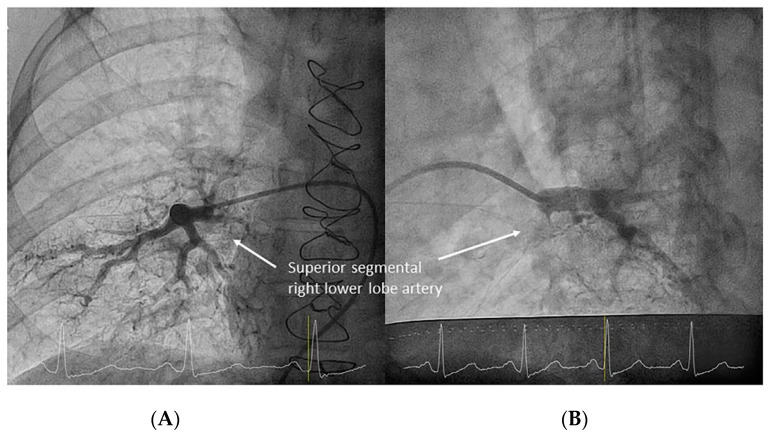
Right lower lobe superior segmental branch (A6). (**A**) anteroposterior view. (**B**) lateral view.

**Figure 18 jcm-10-03358-f018:**
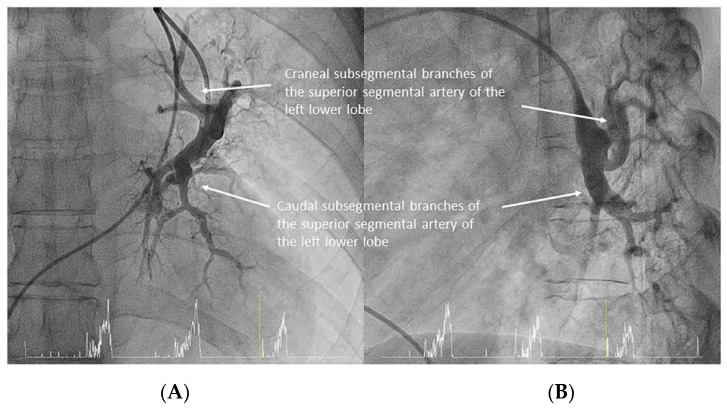
Left lower lobe superior segmental branch (A6): (**A**) anteroposterior and (**B**) lateral views.

**Figure 19 jcm-10-03358-f019:**
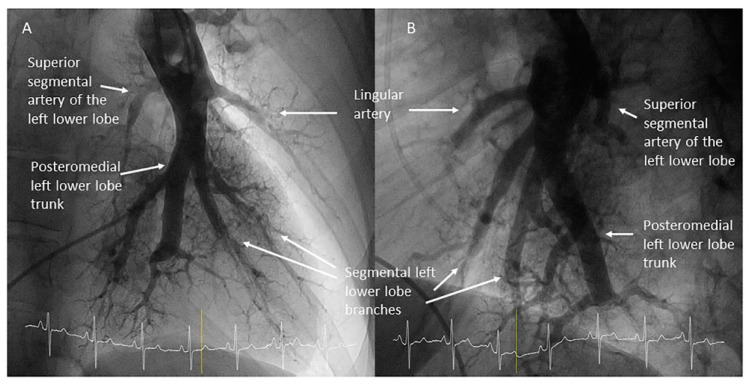
Left lower lobe branches. Posteromedial trunk (A7/10) and two independent basal segmental branches (A8 and A9): (**A**) anteroposterior and (**B**) lateral views.

**Figure 20 jcm-10-03358-f020:**
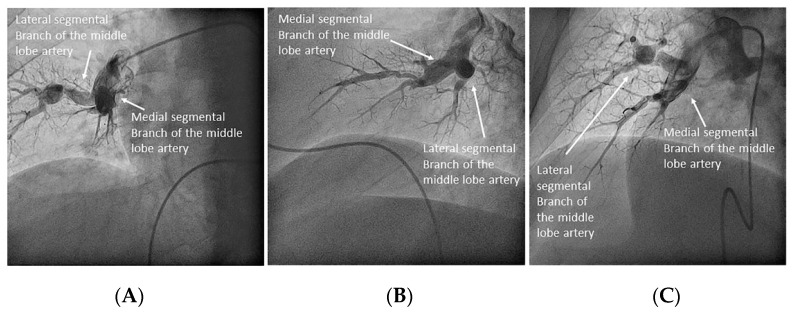
Segmental middle lobe artery branches (A4 and A5): (**A**,**B**) anteroposterior and lateral views overlapping; (**C**) cranial left oblique separates branches.

**Figure 21 jcm-10-03358-f021:**
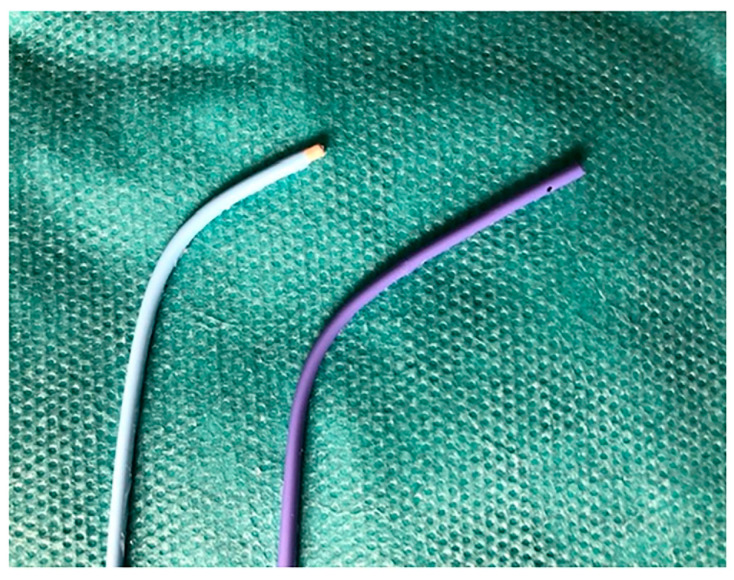
Multipurpose catheters (MP). Blue: MP-A1 (end-hole). Purple: MP-A2 (end-hole + side-holes).

**Figure 22 jcm-10-03358-f022:**
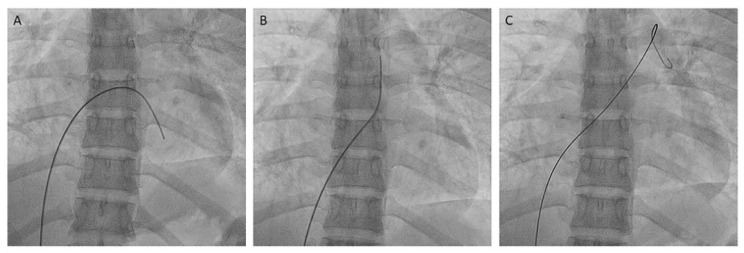
Steps to reach the left pulmonary artery with the MP-A catheter. (**A**): fist step, (**B**): second step, (**C**): third step.

**Figure 23 jcm-10-03358-f023:**
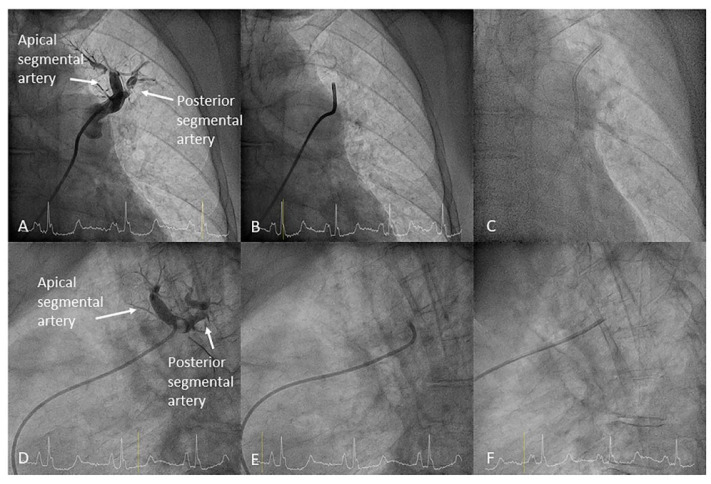
MP catheter positions in the anteroposterior (**A**) (upper row) and lateral views (**B**) (lower row) to cannulate the (**D**,**E**) apical (A1) and (**C**,**F**) posterior (A2) left upper lobe branches.

**Figure 24 jcm-10-03358-f024:**
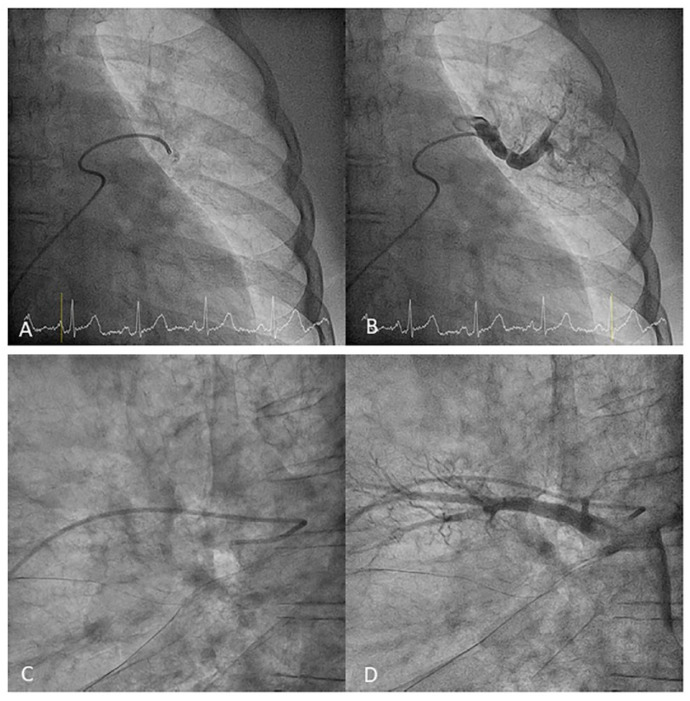
JL catheter position canulating the left upper lobe anterior segmental branch (A3), (**B**,**D**) with and (**A**,**C**) without contrast.

**Figure 25 jcm-10-03358-f025:**
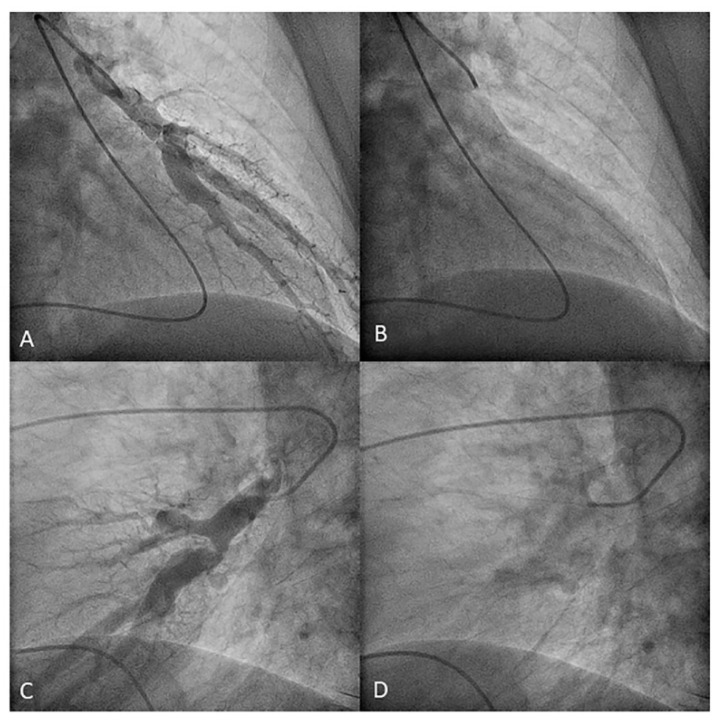
JL catheter position canulating the lingula artery, with (**A**,**C**) and without (**B**,**D**) contrast.

**Figure 26 jcm-10-03358-f026:**
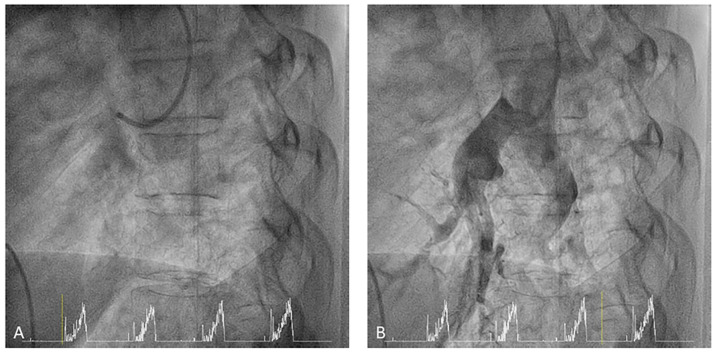
JR catheter position canulating the left lower lobe antero-lateral trunk (A8/9). Lateral view (**A**) without and (**B**) with contrast.

**Figure 27 jcm-10-03358-f027:**
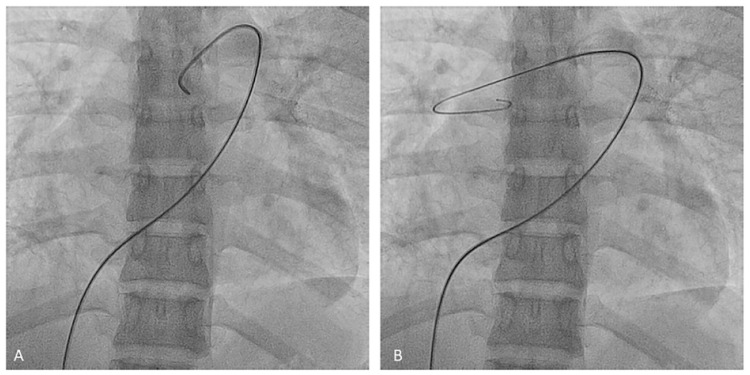
Steps to canulate the right pulmonary artery, (**A**) first step, (**B**) second step.

**Figure 28 jcm-10-03358-f028:**
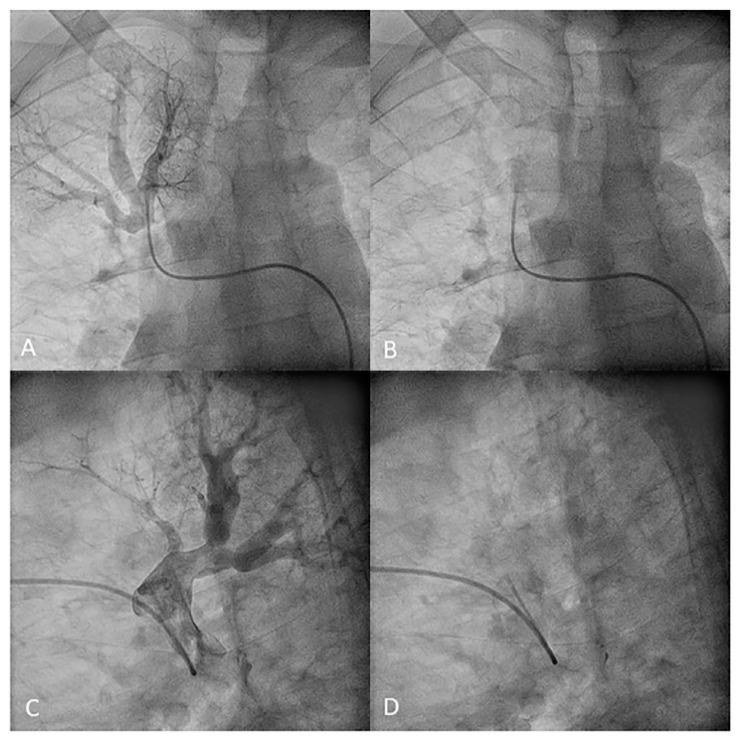
MP catheter position canulating the right upper lobe apico-posterior trunk (A1/2). (**A**,**B**) Anteroposterior view and (**C**,**D**) lateral view.

**Figure 29 jcm-10-03358-f029:**
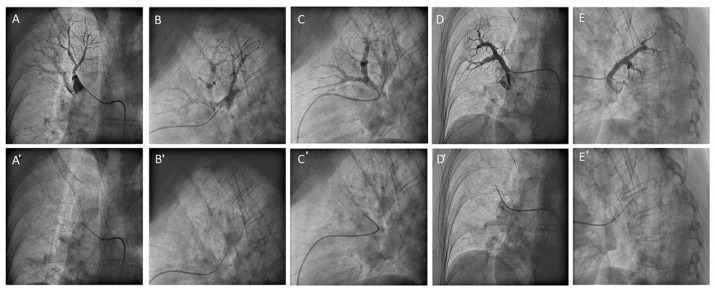
MP catheter canulating the right upper lobe segmental branches, (**A**–**E**) with contrast and (**A’**–**E’**) without contrast. (**A**) Apical branch (A1), anteroposterior view; (**B**) apical branch (A1), lateral view; (**C**) anterior branch (A3), lateral view; (**D**) posterior branch (A2), anteroposterior view; and (**E**) posterior branch (A2), lateral view.

**Figure 30 jcm-10-03358-f030:**
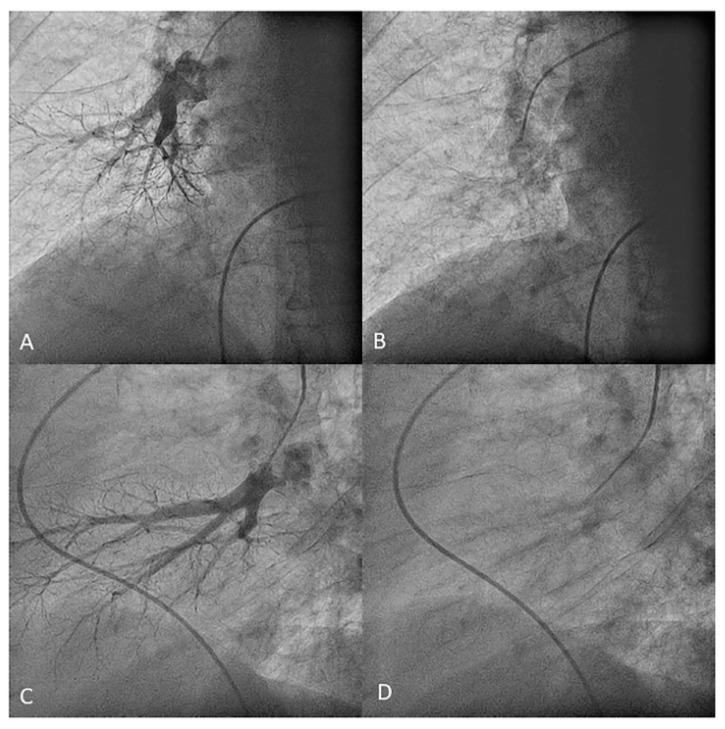
MP catheter canulating the middle lobe artery. (**A**,**B**) Anteroposterior and (**C**,**D**) lateral views.

**Figure 31 jcm-10-03358-f031:**
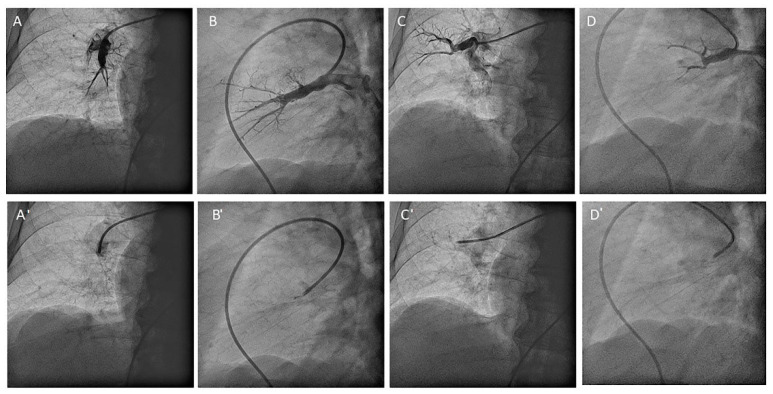
MP catheter canulating the (**A**,**B**) medial (A5) and (**C**,**D**) lateral (A4) middle lobe segmental branches. (**A**,**C**) Anteroposterior and (**B**,**D**) lateral views: (**A**–**D**) with contrast and (**A’**–**D’**) without contrast.

**Figure 32 jcm-10-03358-f032:**
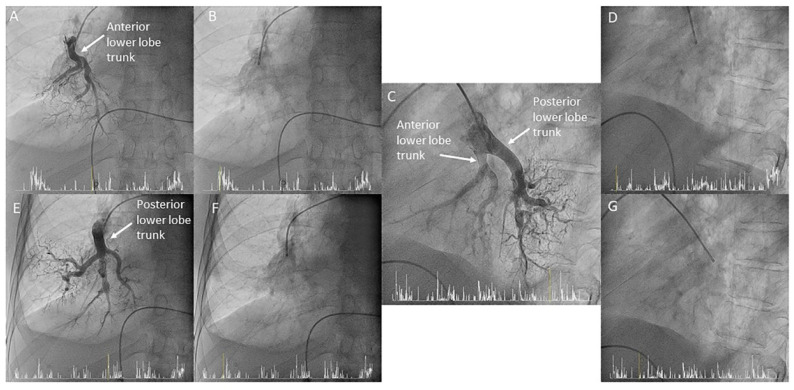
MP catheter canulating the right lower lobe trunks, (**A**,**C**,**E**) with and (**B**,**D**,**F**,**G**) without contrast. (**A**,**B**,**E**,**F**) Anteroposterior and (**C**,**D**,**G**) lateral views.

**Figure 33 jcm-10-03358-f033:**
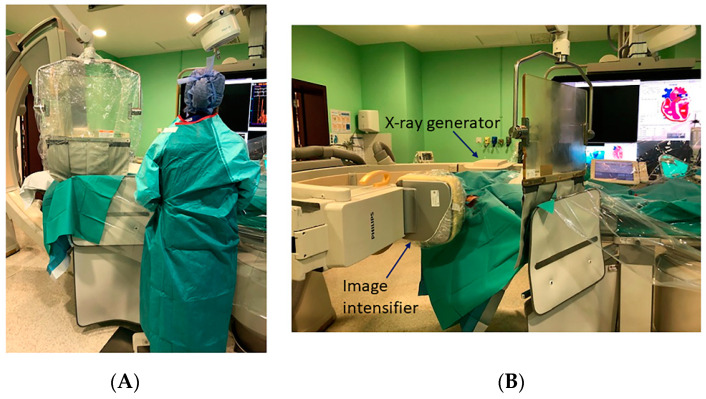
(**A**) Ceiling and table protective shields. (**B**) Reverse lateral projection.

**Table 1 jcm-10-03358-t001:** Procedural and clinical data.

Variable	Value
Age (years)	58.5 ± 15.7
Sex (women)	56.4 (255)
Weight (Kg)	76.9 ± 16.8
Height (cm)	164.5 ± 10.8
Body surface (m^2^)	1.82 ± 0.2
Heart rate (beats per minute)	75.5 ± 13.7
Oxygen saturation (%)	94.1 ± 3.7
Pulmonary artery oxygen saturation (%)	62.5 ± 7.9
Right atrial pressure (mmHg)	8.8 ± 4.6
Systolic pulmonary artery pressure (mmHg)	74.9 ± 22.8
Diastolic pulmonary artery pressure(mmHg)	26.7 ± 13.2
Mean pulmonary artery pressure (mmHg)	43.9 ± 12.8
Wedge pressure (mmHg)	10.8 ± 4.1
Cardiac output (L/min)	3.2 ± 1.5
Cardiac index (L/min/m^2^)	1.8 ± 0.8
Pulmonary vascular resistance (WU)	10.4 ± 4.0
Venous access	
● Femoral	92.0 (416)
● Jugular	5.5 (25)
● Basilica/cephalic/humeral vein	2.4 (11)
Scopia time (min)	19.1 ± 10.6
PDA (Gycm2)	102.3 ± 65.2
Procedure time (min)	81.3 ± 17.9
Contrast dose (mL)	300.8 ± 85.5

Categorical variables: percentage of total procedures (absolute value); continuous quantitative variables: mean ± standard deviation.

**Table 2 jcm-10-03358-t002:** Sub-selective pulmonary angiography complications.

Complication	% (Absolute Value)
Any complication	4.9 (22)
Bleeding	2.2 (10)
Puncture site	1.8 (8)
Hemoptysis	0.2 (1)
Other bleeding	0.2 (1)
Bleeding type (BARC criteria)	
2	1.8 (8)
3	0.4 (2)
Arteriovenous fistula	0 (0)
Pseudoaneurysm	0.7 (3)
Puncture site ischemia	0 (0)
Acute renal failure (AKIN criteria)	1.1 (5)
Needing dialysis	0 (0)
Arrhythmias	0.7 (3)
Atrial fibrillation	0.4 (2)
Paroxysmal Complete AV block	0.2 (1)
Allergic reaction	0.4 (2)
● Cutaneous	0.4 (2)
● Anaphylaxis	0 (0)
● Cardiac tamponade	0 (0)
Respiratory insufficiency requiring ventilatory support	0.2 (1)
Hemodynamic instability requiring pharmacological or mechanical support	0.4 (2)
Death	0.2 (1)

Variables are expressed as a percentage of the total procedures (absolute value).

## Data Availability

Data supporting reported results can be found in the REHAP Spanish National pulmonary arterial hypertension database through mail to rehap@shmedical.es.

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
