# Peer review of "Selective Segmental Pulmonary Angiography: Anatomical, Technical and Safety Aspects of a Must-Learn Technique in Times of Balloon Pulmonary Angioplasty for Chronic Thromboembolic Pulmonary Hypertension"

_jcm, 2021, doi:10.3390/jcm10153358_

Round 1
Reviewer 1 Report
The reviewed article deals with an important clinical aspect related to selective catheterization and selective angiography of segmental pulmonary arteries, which is of particular importance in the era of rapid BPA development. The authors presented the technique of the procedure based on the experience of their own center, and also showed their detailed data on the basic clinical characteristics of the patients and complications. I’ve read the article with great interest.
I have following suggestions to the authors:
- The authors should provide numerical names for pulmonary segmental arteries (left A1, A2, A3, A4, A5, A6, A8, A9, A10, right: A1-A10) and not only descriptive ones. Numerical names are often used in both radiological descriptions and in the literature.
- Other catheters not mentioned by the authors may also be useful in selective angiography of segmental arteries, and especially in BPA (better cath support during intervention). For example, Amplatz Left 1 or 2 for catheterization of the middle lobe arteries of the left pulmonary artery. Authors should mention this in the text of the article.
- Authors should also refer to the types of lesions that can be visualized during selective angiography in CTEPH patients.
Especially since, there are thromboembolic changes visible in a plenty of presented figures. This makes the anatomical assessment difficult in some cases.
Author Response
Please see the attachement

Reviewer 2 Report
The Manuscript "Subselective pulmonary angiography. Anatomical, technical and safety aspects of a must-learn technique in times of balloon 3 pulmonary angioplasty for chronic thromboembolic pulmonary 4 hypertension. " deals with an important topic.
Abstract:
With the advent of balloon pulmonary angioplasty (BPA) for non-surgical chronic throm- 15 boembolic pulmonary hypertension (CTEPH) patients, there is renewed interest in the pulmonary 16 angiography technique. Learning this technique fulfills two main purposes: to identify BPA candi- 17 dates and to provide the operator with the catheter handling needed to perform BPA. -> I think I understand what the authors intend to say - nonetheless, pulmonary angiography is still the standard imaging modality to confirm CTEPH - this is independet of the treatment (medical - surcical - interventional). Therefore, the technique is not limited to the here mentioned main purposes: to identify BPA candiates and to assist the operator; but to diagnose / confirm CTEPH - to evaluate the optimal treatment and so on.
In order to 18 perform a good pulmonary angiography, the operator must know the pulmonary arteries’ anatomy 19 and which are the best angiographic projections and the most suitable catheters to canalize and 20 display each segmental branch. Unfortunately, this information is scarce in the literature. -> of course an IR should know the terretory he treats. Angulations can be assessed from CT or CACT guiding the intervention - additional 3D imaging is very helpful to assess anatomical variants - refer to: Balloon pulmonary angioplasty: applicability of C-Arm CT for procedure guidance.
Eur Radiol. 2016 Nov;26(11):4064-4071. doi: 10.1007/s00330-016-4280-z. Epub 2016 Feb 23. PMID: 26905868 Introduction:Both techniques can identify surgically ac- 39 cessible chronic clots. However, in patients who have only peripheral involvement, none 40 of them is accurate enough to provide fine details of distal vessels(3). -> in the light of CACT or IVUS and so on during the intervention it is questionable how important this is - in the diagnosis it is important to detect CTEPH and to get an idea of the treatment - some details need to be a acquired during BPA procedure.
Instead, a con- 41 ventional invasive sub-selective pulmonary angiography (SSPA) can obtain the best qual- 42 ity images for identifying distal pulmonary thromboembolic disease. -> is this authors opinion or is there some literature ?
an what excactly is meant by "sub-selective pulmonary angiography"
and is the SSPA performed only during BPA or in the forefront ?
the ability to selectively cannulate each segmental branch -> in my opinion this is much easeier with 3D information e.g. CACT and guidance with selective CACT
Figures are good - may be better in DSA technique.
Many times, segmental branches 163 with an anterior distribution are better visualized using cranial and oblique angulations 164 instead of the AP projection. -> true - see the paper mentioned above
We recommend cranial right anterior oblique (RAO) view 165 for the anterior segmental artery of the left upper lobe, the lingula branches and the ante- 166 rior segmental branches of the LLL and cranial left anterior oblique (LAO) view for the 167 anterior segmental artery of the right upper lobe, the middle lobe artery (figure 20) and 168 the anterior segmental branches of the RLL.-> angulations especially craniocaudal or vice versa come to the cost of extended radiation exposure to patients and stuff. I think 3D imaging is at least worth to be mentioned as an alternative here - one could acquire CACT or produce a 2D3D overlay of the conventional CT of the patients - at least for probing of the segmental and sub-segmental arteries the information is very helpful.
of course is an open cava filter no problem...
i still did not get by now what the point of this paper is - is the SSPA described here the standard diagnostic pulmonary angiography or is this the way the authors perform their BPA ?
Standard diagnostic PA would be frontal and lateral projection of each lung for example:
Full text links
Actions
FavoritesShare
Page navigation
- Title & authors
- Abstract
- Similar articles
- Cited by
- Publication types
- MeSH terms
- Substances
- LinkOut - more resources
Comparison of C-arm Computed Tomography and Digital Subtraction Angiography in Patients with Chronic Thromboembolic Pulmonary Hypertension
obviously the authors use a small cardio-detector for imaging and BPA- maybe greater "radiological-detectors" are more suitable.
why dont you cite the more recent 2018 version of the canadian guidelines ? CACT is mentioned there
Practice Guideline Can Respir JDiagnostic evaluation and management of chronic thromboembolic pulmonary hypertension: a clinical practice guideline
Unfortunately, SSAP is not widely available. Expertise 335 is required to perform it, but this is not easy to acquire, as there are no descriptions in the 336 literature on how to perform it. -> in this point again: do the authors intend to do selective PA for diagnosis or during treatment - and i disagree - every physichian who performes BPA must be able to do SSPA.
This image acquisition requires a biplane flat detector, availability 342 of very low magnification tool to include the entire lung in the field of view, digital sub- 343 traction technology and image post-processing. -> to get it in one shot you need a biplane angiography, but you could easily do a frontal and lateral projection. And of course you should do DSA -as it proved to be better compared to non-subtracted images and if you want you have both information on hand subtracted and non-subtracted.
Furthermore, interventional car- 346 diologists (who have assumed in many units CTEPH diagnosis and BPA) are not familiar 347 with the digital subtraction technique nor is this software usually available in their cath- 348 labs. -> so they should familiarize with the technique and should make it available.
Finally, contrast injection 352 through a balloon-tipped catheter in each main pulmonary artery does not provide the 353 operator with the handling of catheters required to perform BPA. -> i dont get the intention of this sentence... why and how (assuming the use of rapid exchange ballons as most colleuge use) should I acquiere images through the ballon-Tipped catheter and not over a guiding catheter ?
Therefore, we suggest 354 the SSPA technique should be learned, because it is a more precise technique than the 355 DSA central angiography to accurately analyze the distal pulmonary arteries involvement, 356 it is more widely available and it facilitates to perform BPA. -> with this sentence the authors imply that others who perform BPA are not able to do this technique - i disagree in this point. moreover the authors did not provide any evidence that SSPA is better compared to DSA of central PA - no comparision was done.
One 382 patient with severe PH and right ventricular disfunction died in the cath-lab due to he- 383 modynamic instability and refractory respiratory failure. -> was this after ballooning for BPA or a vessel injury due to probing for SSPA - if this was due to probing for SSPA I think this is a fatal complication which can be avoided by techniques like CACT.
Learning the SSPA technique is crucial to stablish if a patient with distal CTEPH is a 395 good candidate for BPA and to be able to approach this percutaneous therapy. -> I disagree - every physician who does BPA is able to do SSPA like described here. And there are many other oppurtunities to choose the right patient for BPA not mentioned in this article.
Round 2
Reviewer 2 Report
Thank you for your response.
This technique is still the standard imaging modality to confirm CTEPH which, in addition, helps to stablish the most appropriate treatment. -> I am not sure if "stablished" is the right word in this sence.
Operators interested in perform BPA -> not sure if this is correct think "to perform or in performing"
it is kind of funny that the authors argue with scarce literature and so on to promote their way to do pulmonary angio to do BPA (a seldome procedure compared to cardio angio) but on the ohter hand they state that "
The problem in daily clinical practice is that some of these imaging techniques are not widely available and require expertise" - to my opinion BPA is mostly done in great centers e.g. university hospitals and should not be done in smaller non-specialized clinics - therefore most university hospitals have a radiological cath lab - so why dont cardiologist and radiologist work together in this point ? Moreover, why not gather the expertise in special imaging technique than learning SSPA ?
Thanks for your comment. What I am trying with the figures included in the manuscript is to teach interventional cardiologists the anatomy of the pulmonary arteries through the images they are used to. At the same time, we want to teach them how to cannulate every segmental branch. Our aim is to help them to deal with BPA technique by learning first the diagnostic pulmonary angiography procedure. -> this is a nice aim; nonetheless, i dont believe that every interventional cardiologist should go and treat CTEPH ;) and furthermore, not only cardiologist deal with this disease so do interventional radiologists so I think it should be written more common.
Finally, contrast injection through a balloon-tipped catheter in each main pulmonary artery does not provide the operator with the handling of catheters required to perform BPA. -> i dont get the intention of this sentence... why and how (assuming the use of rapid exchange ballons as most colleuge use) should I acquiere images through the ballon-Tipped catheter and not over a guiding catheter ?
I’ll try to explain this sentence. I mean that performing a pulmonary angiography with central injections in the main pulmonary arteries does not provide the operator with the ability to selectively cannulate each segmental branch. In contrast, performing the SSPA makes her/him canalize selectively each segmental branch with specific catheters and gives him/her the ability to perform BPA once learnt the pulmonary arteries anatomy. -> i get that but waht do you mean with "balloon-Tipped catheter" ? what catheter and how does it work ? we for example use a over the wire balloon and a 6F guide cath performing selective angio through the guide cath - so what does "balloon-Tipped catheter" mean ?
This adverse event happened during a diagnostic SSPA procedure in a patient with CTEPH. Probably the severity of the pulmonary hypertension, right ventricular disfunction, the volume overload and the delay in recognizing the hemodynamic impairment played an important role in the fatal outcome. -> this should be mentioned in the manuscript so the collueges know that even in diagnostic angio fatal complications can happen...
overall nice work.
Author Response
Answer to reviewer 2, second round
-This technique is still the standard imaging modality to confirm CTEPH which, in addition, helps to stablish the most appropriate treatment. -> I am not sure if "stablished" is the right word in this sence.
The word stablish has been changed by “determine” (page 1, line 18)
-Operators interested in perform BPA -> not sure if this is correct think "to perform or in performing"
The sentence has been changed as suggested by this reviewer, “using performing” instead of “to perform” (page 1, line 20)
it is kind of funny that the authors argue with scarce literature and so on to promote their way to do pulmonary angio to do BPA (a seldome procedure compared to cardio angio) but on the ohter hand they state that "
In our center, clinical cardiologists asked interventional cardiologists, years ago, to please perform selective pulmonary angiographies to assess distal involvement in patients with CTEPH, as the central pulmonary angiographies performed by interventional radiologists were not accurate enough. The result was that once interventional cardiologists acquired the ability to move inside the pulmonary vascular tree, for diagnostic procedures, they began performing BPA. Probably, our description may be innovative for countries where these procedures are performed exclusively by interventional radiologists, but our aim is not to promote that interventional cardiologists perform BPA. Our aim is to offer a guide on pulmonary arteries anatomy and a guide on how to perform a selective segmental pulmonary angiography, either for diagnostic or therapeutic purposes, and either for interventional cardiologists or for interventional radiologists.
-The problem in daily clinical practice is that some of these imaging techniques are not widely available and require expertise" - to my opinion BPA is mostly done in great centers e.g. university hospitals and should not be done in smaller non-specialized clinics - therefore most university hospitals have a radiological cath lab - so why don’t cardiologist and radiologist work together in this point ? Moreover, why not gather the expertise in special imaging technique than learning SSPA ?
We share the opinion that BPA should be done in great centers. In fact, our center is a University Hospital with an Expert Pulmonary Hypertension Unit which perform TEA since 2000 and BPA since 2013. The ideal situation would be that this procedure is performed only in this kind of centers. However, the reality, at least in our country, is that interventional cardiology units of many referral centers are interested in begin BPA and their operators are asking for help to learn how to do it.
In our country, most interventional radiologists are not interested in this field or, at least, they haven’t shown interest in it by the moment, whilst the situation is just the opposite between interventional cardiologists. Right heart catheterizations and pulmonary angiographies in patients with CTEPH are performed mostly in cardiologic cath-labs.
When I began performing selective segmental pulmonary angiography and, years later, BPA, I had to learnt by myself how to canalize every segmental pulmonary branch, as this information was not detailed in the literature. I would have liked having a guide like the one written by us when I began performing pulmonary angiography and BPA. Thus, I think it will be useful for novel operators interested in this field to have a manuscript where they can consult how to perform a selective segmental pulmonary angiography.
Nowadays, as interventional cardiologists are asking for help, we are teaching them the theoretical aspects and the practical aspects of BPA, organizing courses in our center and helping them beginning these programs in their centers. Our aim is to increase safety for patients, because as you know, there is no low forbidding them to perform BPA. That’s why, once assumed the reality, which is that non expert centers will perform BPA, we think it is more convenient for them and their patients to work together with them and teach them how to perform diagnostic and interventional procedures in these patients.
Of course, it is very important gathering the expertise in special imaging technique, but in our opinion, learning SSPA is also worth it, as forces the operator to learn the anatomy of the pulmonary arteries, which are the best catheters to cannulate every one and which projections are the best to display every segmental branch, and all these skills are really useful to, posteriorly, perform BPA. One thing does not exclude the other.
Thanks for your comment. What I am trying with the figures included in the manuscript is to teach interventional cardiologists the anatomy of the pulmonary arteries through the images they are used to. At the same time, we want to teach them how to cannulate every segmental branch. Our aim is to help them to deal with BPA technique by learning first the diagnostic pulmonary angiography procedure. -> this is a nice aim; nonetheless, I dont believe that every interventional cardiologist should go and treat CTEPH ;) and furthermore, not only cardiologist deal with this disease so do interventional radiologists so I think it should be written more common.
I agree with this opinion. Not all interventional cardiologists should treat CTEPH, in fact it is a minority of them who are interested in this field. This minority is the group of operators our manuscript is aimed at. It is true that in many European countries interventional radiologists are the operators who deal with this disease, as this reviewer points out. So, the sentence has been changed, to be more inclusive.
Final sentence: “For these reasons we propose that interventional cardiologists or interventional radiologists who want to perform BPA should first of all learn to carry out SSPA diagnostic procedures, following the technique explained in this manuscript.” Page 19, lines 373-376
Finally, contrast injection through a balloon-tipped catheter in each main pulmonary artery does not provide the operator with the handling of catheters required to perform BPA. -> I dont get the intention of this sentence... why and how (assuming the use of rapid exchange balloons as most collegue use) should I acquire images through the balloon-Tipped catheter and not over a guiding catheter?
During the BPA procedure, images are acquired through the guiding catheter of course. What I mean is that the San Diego’s group diagnostic DSA pulmonary angiography technique, which performs the pulmonary angiography through a central injection with a balloon tipped catheter (like Berman catheter) in each main pulmonary artery, does not provide the operator with the ability of cannulating every segmental branch. In contrast, the diagnostic selective segmental pulmonary angiography we explain in this manuscript requires that the operator cannulates every segmental branch, and thus, provides him with this ability.
Please find below the final sentence in the paragraph which remarks the differences between diagnostic DSA central pulmonary angiography and selective segmental pulmonary angiography, on page 19, lines 370-373:
“Finally, contrast injection through a balloon-tipped catheter in each main pulmonary artery with diagnostic purposes, does not provide novel operators with the handling of catheters required to canalize independently each segmental branch, which will be needed to perform BPA”
I’ll try to explain this sentence. I mean that performing a pulmonary angiography with central injections in the main pulmonary arteries does not provide the operator with the ability to selectively cannulate each segmental branch. In contrast, performing the SSPA makes her/him canalize selectively each segmental branch with specific catheters and gives him/her the ability to perform BPA once learnt the pulmonary arteries anatomy. -> i get that but waht do you mean with "balloon-Tipped catheter" ? what catheter and how does it work ? we for example use an over the wire balloon and a 6F guide cath performing selective angio through the guide cath - so what does "balloon-Tipped catheter" mean ?
The balloon tipped catheter is the one used for performing the diagnostic San Diego’s group DSA pulmonary angiography technique,,i.e. Berman catheter. These are the balloon tipped catheters we refer to. We don’t use balloon tipped catheter, neither for the diagnostic procedure nor for the BPA procedure. As it is explained in the text, we use diagnostic catheters (non balloon-tipped) for the diagnostic procedure and guide catheters for the BPA procedure. We use the same guidewires we use for coronary angioplasty (0,014 inches) and monorail balloons, which, opposite to rapid exchange balloons, do not require long guidewires.
This adverse event happened during a diagnostic SSPA procedure in a patient with CTEPH. Probably the severity of the pulmonary hypertension, right ventricular dysfunction, the volume overload and the delay in recognizing the hemodynamic impairment played an important role in the fatal outcome. -> this should be mentioned in the manuscript so the colleges know that even in diagnostic angio fatal complications can happen...
I have added this information in the manuscript, to make operators aware of the fragile hemodynamic situation of these patients.
Please see below the final paragraph:
“One patient with severe PH and right ventricular dysfunction died in the cath-lab due to hemodynamic instability and refractory respiratory failure. Probably the PH severity, the right ventricular dysfunction, the volume overload and the delay in recognizing the hemodynamic impairment played an important role in the fatal outcome. Therefore, although operators must be aware of the fragility of these patients, our peri-procedural mortality rate of 0.2% confirms the safety of the SSPA technique in an expert PH center.” (Page 20, lines 419-425)
Please see the attachment

This manuscript is a resubmission of an earlier submission. The following is a list of the peer review reports and author responses from that submission.